# Learning from All: Concept Alignment for Autonomous Distillation from Multiple Drifting MLLMs

## Abstract

This paper identifies a critical yet underexplored challenge in distilling from multimodal large language models (MLLMs): the reasoning trajectories generated by multiple drifting teachers exhibit concept drift, whereby their reasoning distributions evolve unpredictably and transmit biases to the student model, ultimately compromising its performance. To tackle this issue, we pioneer a theoretical connection between concept drift and knowledge distillation, casting the non-stationary reasoning dynamics from multiple MLLM teachers as next-token prediction of multi-stream reasoning trajectories. Guided by concept drift, we introduce the "learn–compare–critique" paradigm, culminating in autonomous preference optimization (APO). Under the active guidance of the teachers, the student model first learns and self-distils preferred thinking by comparing multiple teachers. It then engages in critical reflection over the drifting inference from teachers, performing concept alignment through APO, ultimately yielding a robust, consistent, and generalizable model. Extensive experiments demonstrate our superior performance of consistency, robustness and generalization within knowledge distillation. Besides, we also contributed a large-scale dataset CXR-MAX (Multi-teachers Alignment X-rays), comprising 170,982 distilled reasoning trajectories derived from publicly accessible MLLMs based on MIMIC-CXR. Our code and data are public at: `https://anonymous.4open.science/r/Autonomous-Distillation/`.

## 1 Introduction

Knowledge distillation (KD) has emerged as a central paradigm for transferring knowledge from general large-scale teacher models, such as multi-modal large language models (MLLMs), to more compact student models. This is particularly in customized domain-sensitive settings such as the medical field Shen et al. (2025a); Shu et al. (2025); Cao et al. (2025a); Feng et al. (2025b). Recent advances in multi-teacher distillation further highlight the promise of leveraging complementary expertise and diverse domain-specific priors from multiple teacher models to enhance the student's learning Gu et al. (2025a); Chen et al. (2024a). However, this paradigm remains fundamentally constrained by the non-stationary dynamics of drifting MLLMs. Specifically, the inherent *inter-model drift* among multiple teachers progressively destabilizes the *concept alignment* between the student model's representation space and the real-world domain. This misalignment driven by model drift can lead to catastrophic error propagation, critically threatening the reliability of models in safety-sensitive applications.

Concept drift theory offers a compelling analytical lens to examine concept alignment. It allows us to characterize not only the *inter-model drift* among MLLM teachers but also the *hierarchical drift* between the teacher ensemble and the student model under non-stationary custom-tuning Lu et al. (2019); Yang et al. (2025a). This framework is adept at capturing the unpredictable distributional shifts that unfold across their parallel reasoning trajectories. Within this perspective, the autoregressive decoding paradigm of a single MLLM can be viewed as a sequential token-generation process, where each step propagates through the model's latent reasoning pathways. Consequently, when multiple teachers are involved, their parallel autoregressive processes are transformed into a multi-stream drift scenario, where each process follows its own distributional dynamics that may drift

asynchronously and in heterogeneous directions. These asynchronous drifts, which can manifest as convergence, divergence, or even direct conflict, fundamentally distinguish the student's learning challenge from the single-teacher setting.

Within the concept drift framework, our analysis uncovers fundamental limitations in distilling knowledge from multiple drifting MLLMs in customized domain-specific scenarios. Figure 1a presents the diagnostic reasoning outcomes of various teacher MLLMs on MIMIC-CXR, featuring seven leading publicly accessible MLLMs with precision, recall, and semantic similarity, where precision and recall specifically refer to the alignment between radiological findings in the MLLM generated reports and in the ground-truth radiology reports written by radiologists, rather than disease-classification accuracy. The limitations of multiple drift MLLMs in domain-specific scenarios are manifested in:

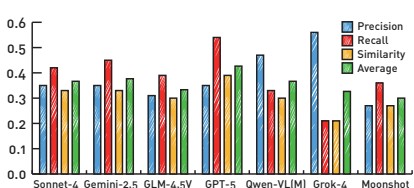

(a) Concept Drift Among MLLM Teachers

**Distilled Student**

Based on the provided PA chest X-ray, the following key findings support the diagnoses of atelectasis and pleural effusion:
1. Atelectasis:

- Volume Loss in the Left Lower Lung Field: The left lower lung appears significantly darker than the right, indicating volume loss.
  # Inconsistency: atelectasis is denser (whiter), not darker.
  …

2. Pleural Effusion:

- Bilateral Pleural Opacities: Both lungs show areas of increased opacity, but they are more pronounced on the left.
  # Overgeneralization: bilateral opacities may have multiple causes and cannot be considered the primary cause.
- Possible Pleural Line: There may be a pleural line visible along the midline, indicating pleural fluid accumulation.
  # Inconsistency: pleural line along the midline usually indicates pneumothorax, not pleural effusion.
- No Evidence of Pulmonary Infiltrates: The lung fields appear relatively clear, with no signs of pulmonary infiltrates, consolidation, or ground-glass opacities.
  # Overconfident: infiltrates may be masked.
  …

**Observation 1.1** *Notable divergence and complementarity exists among teacher models: while Qwen-VL-Max and Grok-4 achieve high precision but relatively low recall, favoring concise yet accurate feature descriptions. GPT-5 attains the highest recall with less impressive precision, capturing a broader range of imaging features at the cost of introducing potentially redundant details. This highlights that the substantial concept drift inherent among teachers poses a significant challenge for distillation, yet their pronounced complementarity presents a compelling advantage over distillation from any individual teacher.*

**Observation 1.2** *Distilled student also inherits distributional biases from drifting MLLM teachers: In Figure 1b, the distilled student does not merely inherit the complementary strengths of multiple drifting MLLMs but also internalizes their distributional biases, leading to systematic errors such as overgeneralization, semantic inconsistency, and erroneous rationalization. As a result, the student becomes a drift-biased amalgamation of its teachers rather than a balanced integrator.*

(b) Example of Drift-Biased Distilled Student. Red markers pinpoint specific flaws in the generated reports, with green markers providing the rationale behind these errors.

Figure 1: Transmission of Concept Drift behind Distillation of MLLMs

Therefore, summarizing the above challenges of knowledge distillation from multiple drifting MLLMs, it raises the important question:

*How to maintain concept alignment in distillation with multiple drifting MLLM teachers?*

Inspired by advances in reinforcement learning Yang et al. (2025b); Rafailov et al. (2023a); Agarwal et al. (2024), we propose the autonomous distillation that leverages multiple drifting MLLM teachers to build a generalized, consistent and robust student model under customized specific domain.

It follows the principled "learn–compare–critique" paradigm. First, in the learning stage, the student model absorbs knowledge distilled from multiple teachers, thereby acquiring broad domain-specific expertise while inevitably inheriting certain biases. Then, in the comparison stage, the student performs self-distillation by aligning concepts across the outputs of different teachers, which refines the acquired knowledge into a more consistent representation. Finally, in the criticism stage, we introduce autonomous preference optimization (APO), where the consensus knowledge distilled through self-alignment is treated as a preferred preference, while the biased components of individual teachers are regarded as negative preferences. It leverages the reinforced learning to mitigate biases inherited from drifting teachers in the initial learning stage, enabling effective and efficient preference alignment that enhances robustness and generalization.

In summary, our paper mainly makes the following contributions:

1. We establish a novel theoretical framework that casts the autoregressive generation of reasoning trajectories in MLLMs within the perspective of concept drift theory, providing a principled foundation for understanding and analyzing knowledge distillation from multiple drifting teachers.

2. Second, building upon the concept drift theory, we design an autonomous distillation framework following a "learn–compare–critique" paradigm. The student first absorbs broad domain-specific knowledge from diverse teachers, then conducts self-distillation to align their concepts, and ultimately employs autonomous preference optimization to reconcile biases and reinforce generalization. Unlike traditional knowledge distillation, the drifting teachers here also act as negative signals, which helps approximate and refine the decision space of the student model.

3. Third, we conduct comprehensive empirical evaluations on diverse clinical benchmarks for chest radiographs, including disease classification, diagnostic report generation, and zero-shot generalization. Despite relying on only one-tenth of the distillation data typically required, our method consistently delivers statistically significant gains in robustness, generalization, and accuracy under multiple drifting MLLM teachers. Furthermore, ablation studies substantiate the effectiveness of each component of our framework.

4. As a pioneer contribution to the community, we introduce CXR-MAX (**M**ulti-teahers **A**lignment **X**-rays), a large-scale dataset comprising 170,982 distillation results of reasoning trajectories derived from publicly accessible MLLMs based on MIMIC-CXR.

## 2 METHODOLOGY

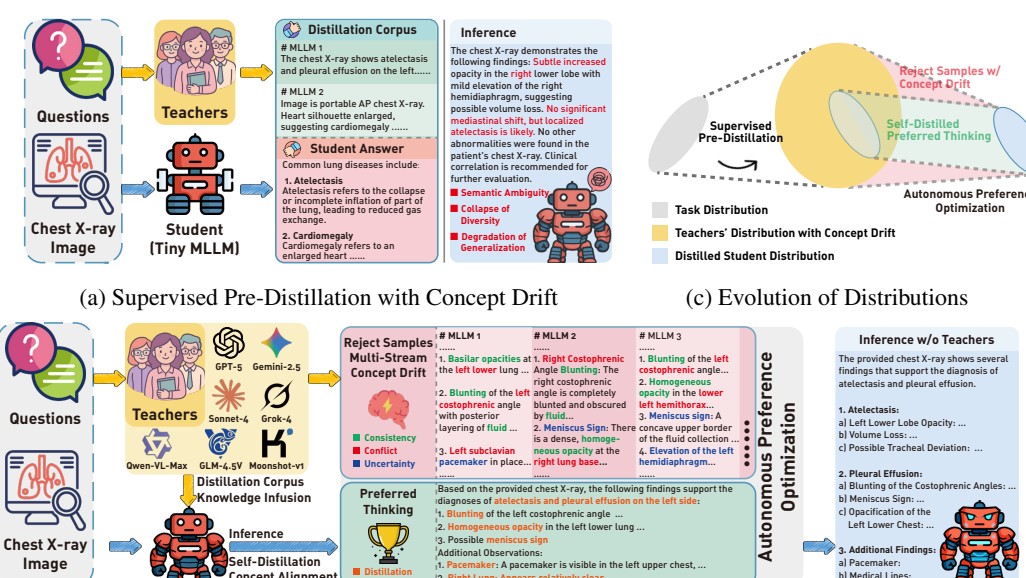

(a) Supervised Pre-Distillation with Concept Drift      (c) Evolution of Distributions

(b) Autonomous Distillation from Multiple Drifting Teachers

Figure 2: The main contributions of our methods. (a) By formalizing the autoregressive inference of MLLM teachers as multi-stream next-token prediction under the lens of concept drift, we reveal that inter-teachers' disturbances of reasoning can propagate to the student via supervised pre-distillation, inducing unpredictable drifts. (b) We propose autonomous preference optimization (APO), leveraging reasoning trajectories carrying explicit conflicts and uncertainties from drifting MLLMs as negative samples, whereas crystallized thinkings via self-distillation as positive signals. Driven by reinforced learning, our model follows a "learn–compare–critique" paradigm to autonomously perform preference alignment, yielding a more robust and generalizable domain-specialized student model. (c) The distribution evolution at different stages is exhibited. Multiple teacher models first map the task distribution to a student-amenable space; the student then learns from this space while simultaneously reflecting on inter-teacher drift, thus autonomously refining itself.

## 2.1 Understanding Teacher Divergences with Multi-Stream Concept Drift

In this section, we extend the notion of concept drift to the setting of multi-teacher MLLMs, emphasizing the unpredictable distributional shifts that arise across the reasoning trajectories of different teachers. Prior studies on concept drift predominantly address single-stream inference Yang et al. (2025b), where an individual MLLM $\pi$ autoregressively generates the token at position $j$ in the reasoning chain by recursively performing on-policy sampling conditioned on the visual input $v$ and textual prompt $l$, such that the partial token sequence $t_{<j}$ of the CoT trajectory is given by

$$t_j \sim \pi(\cdot \mid v, l, t_{<j}). \tag{1}$$

Accordingly, the single-stream reasoning process of an MLLM can be formalized as follows:

**Definition 2.1** *Intuitively, the autoregressive reasoning trajectory of an MLLM unfolds as a chain-of-thought (CoT) stream $S_{0,i} = \{s_0, \ldots, s_i\}$, where each state $s_j = (t_{<j}, z_j)$ comprises the partial token sequence $t_{<j}$ generated up to step $j$ together with the corresponding latent predictive distribution $z_j$ that governs the subsequent generation.*

Capitalizing this formulation, we generalize the concept drift framework from a single MLLM to the multi-teacher setting, where each teacher is associated with a distinct reasoning trajectory. Formally, we define the multi-stream concept drift of the reasoning process as follows:

**Definition 2.2** *Consider $N$ CoT streams corresponding to $N$ MLLM teachers, with their reasoning trajectories denoted by $\mathbb{S}_{0,i} = \{\mathcal{S}_0, \ldots, \mathcal{S}_i\}$, where $\mathcal{S}_j = (s_j^1, \ldots, s_j^N)$, and $s_j^m$ represents the state from the $m$-th teacher. Let $\mathbb{S}_{0,i}$ follow a certain distribution $F_{0,i}(\mathcal{S}m)$. Concept drift among the various MLLM teachers is said to occur at step $m + 1$ if $P_{0,i}(\mathcal{S}_m) \neq P_{i+1,\infty}(\mathcal{S}_m)$, which can be expressed as*

$$\exists t : P_i(\mathcal{S}_m) \neq P_{i+1}(\mathcal{S}_m). \tag{2}$$

*where the teachers' predictions are conditionally independent given their own prior states, leading the joint probability $P_i(\mathcal{S}_m)$ can be decomposed as*

$$P_i(\mathcal{S}_m) = P_i(t_{<m}^1, \ldots, t_{<m}^N) \prod_{u=1}^{N} P_i(z_m^u | t_{<m}^u) \tag{3}$$

Furthermore, we assume that the teachers do not interact during reasoning, so their historical trajectories $t_{<m}$ are statistically independent, allowing the joint distribution to be factorized into the product of the marginal distributions of individual teachers, thereby yielding a fully decomposed form:

$$P_i(\mathcal{S}_m) = \prod_{u=1}^{N} P_i(t_{<m}^u) P_i(z_m^u | t_{<m}^u) \tag{4}$$

Therefore, with the notion of multi-stream concept drift of MLLM reasoning trajectories in place, we are able to effectively capture the dynamic variations that emerge during the knowledge distillation from multiple teachers, formalized as the concept drift process $\prod_{u=1}^{N} P_i(t_{<m}^u)$, and its induced probabilistic divergence $\prod_{u=1}^{N} P_i(z_m^u | t_{<m}^u)$,. Specifically, the framework enables us to quantify the evolving discrepancy between the expected predictive distribution and the realized distribution of cognitive states throughout the reasoning trajectory, which directly governs the alignment of concepts learned by the student.

## 2.2 Harmonizing Drifting MLLM Teachers via Concept Alignment

Building on the formalization of concept drift across multiple MLLM teachers in Eq. 4, we reveal intrinsic inconsistencies and biases in their domain knowledge that, if distilled naively, would inevitably propagate to the student model and manifest as systematic errors, as demonstrated in Observation 1.2. To address this challenge, we propose an approach that integrates the drifting knowledge of multiple teachers while enforcing concept alignment not only across teachers but also between the teachers and the student model through self-distillation.

First, our model undergoes a supervised pre-distillation phase with multiple drifting MLLM teachers, during which it broadly assimilates their collective predictions despite the presence of concept drift, as illustrated in Fig. 2a. Specifically, at each reasoning step $m$, the teachers provide a set of predictive distributions $\mathcal{Z}_m$ from which the student model absorbs heterogeneous knowledge across distinct reasoning trajectories, formalized as

$$\mathcal{Z}_m = \{P_i(z_m^u|t_{<m}^u)\}_{u=1}^N \tag{5}$$

Thus, in this stage, the goal of the student is to project domain-specific distributions into the representational concept space jointly recognized by the MLLM teachers, as depicted in Fig. 2b. Formally, the objective can be expressed as

$$q^*(z_m|t_{<m}) = \arg\min_q \sum_{u=1}^N D_{KL}(P_i(z_m^u|t_{<m}^u)||q(z_m|t_{<m})) \tag{6}$$

where $q^*$ denotes the unified distribution that encapsulates the collective knowledge of all MLLM teachers within the student model. In this context, the student integrates the drifting teacher distributions into a coherent internal representation, reconciling heterogeneous signals not by following any single teacher, but by minimizing their collective divergence from an aligned target distribution $q(z_m \mid t_{<m})$.

Consequently, the pre-distilled student model $\hat{\pi}_{st}$ has assimilated the specific-domain knowledge from diverse teachers via pre-distillation, mitigating potential omissions during the reasoning. The subsequent step addresses the drift across teachers by employing self-distillation to compare their underlying concepts and extract the consistent ones. Specifically, We aggregate the CoT sequences $\mathcal{T} = \{t^1, \ldots, t^N\}$ generated by various teachers for the same instance, and condition the sampling process on the concatenated trajectories jointly with the input image $v$ and prompt $l$, thereby deriving self-distilled outcomes that enforce conceptual alignment:

$$t^+ \sim \hat{\pi}_{st}(\cdot|v, l, \mathcal{T}) \tag{7}$$

At this stage, our framework consolidates insights from multiple drifting MLLM teachers through self-distillation, yielding the preferred reasoning trajectories while distilling the teachers' feature space into the student.

## 2.3 Aligning Distilled Concepts via Autonomous Preference Optimization

Beyond relying on teachers' guidance $\mathcal{T}$ to derive preferred reasoning trajectories $t^+$ in Eq.7, the target student model is expected to autonomously refine its reasoning, toward independent, self-consistent and generalized conclusions. Consequently, following the "learn-compare-critique" paradigm, the bias reasoning trajectories $\mathcal{T}$ from drifting MLLM teachers serve as negative signals, prompting the student model to critically re-evaluate and refine its acquired knowledge. Through conceptual alignment, the student identifies coherent and rational explanations that resolve inconsistencies, thereby enabling iterative self-improvement and evolutionary learning.

Formally, drawing inspiration from DPO Rafailov et al. (2023b), we derive the optimal policy that maximizes the reward function:

$$r(v, l, t) = \beta \log \frac{\pi_\theta(t|v, l)}{\hat{\pi}_{st}(t|v, l)} \tag{8}$$

where $\beta$ is a parameter controlling the deviation from the base reference policy $\hat{\pi}_{st}$, namely the pre-distilled student model, and $\pi_\theta$ denotes the autonomous refining model. With the reward function, we treat the distilled reasoning trajectories $t^+$ in Eq.7 as the preferred thinking, whereas the raw outputs $\mathcal{T}$ from drifting MLLM teachers are regarded as negative signals that provide biased guidance. Thus, based on the Bradley-Terry model Bradley & Terry (1952), the preferred distilled distribution can be written as:

$$P(t^+ \succ \mathcal{T}|v,l) = \frac{\exp(r(v,l,t^+))}{\exp(r(v,l,t^+)) + \sum_{u=1}^{N} w_u \exp(r(v,l,t^u))} \tag{9}$$

Having expressed the probability of distilled preference reasoning trajectories in terms of the optimal policy, we can now formulate a maximum likelihood objective for the parametrized policy $\pi_\theta$, which objective is given by:

$$\mathcal{L}_{APO} = -\mathbb{E}_{(v,l,t^+,\mathcal{T})} \left[ \log P(t^+ \succ \mathcal{T}|v,l) \right] \tag{10}$$

In this context, combinted with Eq.8, Eq.9 and Eq.10, the antonomous preference optimization (APO) is driving the reinforced distillation with multiple drifting MLLMs teachers through the maximum likelihood objective:

$$\mathcal{L}_{APO} = -\mathbb{E}_{(v,l,t^+,\mathcal{T})} \left[ \log \frac{(\frac{\pi_\theta(t^+|v,l)}{\hat{\pi}_{st}(t^+|v,l)})^\beta}{(\frac{\pi_\theta(t^+|v,l)}{\hat{\pi}_{st}(t^+|v,l)})^\beta + \sum_{u=1}^{N} w_u (\frac{\pi_\theta(t^u|v,l)}{\hat{\pi}_{st}(t^u|v,l)})^\beta} \right] \tag{11}$$

Against this backdrop, by adhering to the paradigm of "learning-compare-critique", the student model attains concept alignment under the influence of drifting MLLM teachers, thereby mitigating the transmission of biased knowledge. This adaptive evolution not only enhances the model's internal consistency but also strengthens its generalization and robustness, laying a principled foundation for reliable reasoning across diverse domains.

## 2.4 BUILDING CXR-MAX DATASET FOR MULTI-TEACHERS REASONING

Since we are pioneers in introducing the concept drift into the knowledge distillation of multiple MLLMs, we are deeply aware of the scarcity of multiple CoT from various MLLMs in downstream tasks, especially in the highly professional medical field. Consequently, we aim for the model to autonomously adapt to concept drift, selectively assimilating consistent and valuable knowledge from multiple teachers while preventing the inheritance of biases during distillation.

In this context, to rigorously evaluate the potential of a student model trained under multiple drifting teachers, a more realistic training dataset for knowledge distillation is essential. Addressing the need for high-quality chain-of-thought (CoT) data from diverse MLLMs, we introduce CXR-MAX (**M**ulti-teachers **A**lignment for **X**-rays), an extension of the MIMIC-CXR dataset Johnson et al. (2019) incorporating outputs from seven widely used public MLLMs. CXR-MAX provides 170,982 distillation instances of reasoning trajectories covering 14 thoracic pathologies, establishing the first large-scale benchmark for knowledge distillation with multiple MLLM teachers' reasoning trajectories in clinical chest X-ray interpretation. Additional details are provided in Appendix B.

## 3 EXPERIMENTS

In this section, we verify the robustness, consistency and generalization of our proposed autonomous distillation under multiple drifting teachers.

The MIMIC-CXR dataset Johnson et al. (2019) serves as an ideal training environment for our method, since medical diagnosis embodies the sophisticated reasoning and high-stakes practicality that our distillation approach aims to capture. It presents 371,920 chest X-rays associated with 227,943 imaging studies from 65,079 patients. And images are provided with 14 labels with corresponding free-text radiology reports, namely Atelectasis (Ate.), Cardiomegaly (Car.), Consolidation (Con.), Edema (Ede.), Enlarged Cardiomediastinum (ECM), Fracture (Fra.), Lung Lesion (LL), Lung Opacity (LO), Pleural Effusion (PE), Pneumonia (Pna.), Pneumothorax (Pnx.), Pleural Other (PO), Support Devices (SD) and No Finding (NF).

Acknowledging the additional computational overhead and costs associated with employing multiple teachers, we intentionally and deliberately restricted our distillation to only 1/10 of the whole MIMIC-CXR, underscoring the efficacy of our method in achieving high-quality knowledge transfer

from the drifting teachers, even under limited data conditions. The list of chosen random samples is given in our code.

Additionally, we relied solely on the classification labels from MIMIC-CXR and did not utilize the original radiology reports for training. It is motivated by our focus on knowledge transfer from MLLMs as teachers, as well as the limited generalizability of human-annotated reports with reasoning trajectories, which are often scarce in the domain-specific area.

In terms of the model, we employ Qwen2.5-VL (7B) Bai et al. (2025) as the student model to perform supervised pre-distillation and autonomous preference optimization, cascadedly. And they only train one epoch for each stage with a batch size of 2. More detailed experimental implementations are given in Appendix C.

| | Venue | Con. | PE | Pna. | Pnx. | Ede. | Avg. |
|---|---|---|---|---|---|---|---|
| *Full Data Training* | | | | | | | |
| CTrans | CVPR'23 | 0.44 | 0.61 | 0.45 | 0.32 | 0.66 | 0.49 |
| CheXRelNet | MICCAI'22 | 0.47 | 0.47 | 0.47 | 0.36 | 0.49 | 0.45 |
| BioViL | ECCV'22 | 0.56 | 0.63 | 0.60 | 0.43 | 0.68 | 0.58 |
| BioViL-T | CVPR'23 | 0.61 | 0.67 | 0.62 | 0.43 | 0.69 | 0.60 |
| Med-ST | ICML'24 | 0.61 | 0.67 | 0.59 | 0.65 | 0.54 | 0.61 |
| TempA-VLP | WACV'25 | 0.65 | 0.59 | 0.73 | 0.43 | 0.77 | 0.64 |
| CoCa-CXR | Arxiv'25 | 0.70 | 0.70 | 0.61 | 0.73 | 0.72 | 0.69 |
| *10% Data Training w/o Radiologist Reports* | | | | | | | |
| **Ours** | **This paper** | **0.84** | **0.67** | **0.78** | **0.96** | **0.65** | **0.78** |

Table 1: **Evaluation results of multi-label chest diseases classification on MS-CXR-T.** Top-1 accuracy is applied to evaluate the performance of different methods. The best-performing models are highlighted in red, with the second-best in blue. Comparison methods include CTrans Bannur et al. (2023b), CheXRelNet Karwande et al. (2022), BioViL Boecking et al. (2022) , BioViL-T Bannur et al. (2023b) , Med-ST Yang et al. (2024a) ,TempA-VLP Yang & Shen (2025) and CoCa-CXR Chen et al. (2025).

## 3.1 ORCHESTRATING MULTIPLE DRIFTING TEACHERS FOR ROBUST MLLM DISTILLATION

First, to explicitly demonstrate the superior performance of our proposed method under non-stationary environments, especially in robustness, we compare it with other models on MS-CXR-T Bannur et al. (2023a), where instances are chosen from the public MIMIC-CXR. As presented in Table 1, our autonomous distillation approach attains a superior overall performance of 0.76, outperforming the second-best method, CoCa-CXR Chen et al. (2025) by nearly 13%, especially for only 1/10 of the data without radiologist reports. This result underscores the robustness of our framework under instruction from multiple drifting MLLMs. While our method trails the top-performing CoCa-CXR by a narrow margin of 0.03 on pleural effusion (PE), we attribute this performance gap to CoCa-CXR's use of additional data from Chest ImaGenome Wu et al. (2021) in addition to stan-

| MLLMs | Con. | PE | Pna. | Pnx. | Ede. | Avg. |
|---|---|---|---|---|---|---|
| *MLLM Teachers with Huge Parameters* | | | | | | |
| GPT-5 | 0.75 | 0.68 | 0.89 | 0.90 | 0.52 | 0.75 |
| Gemini-2.5 | 0.28 | 0.61 | 0.40 | 0.94 | 0.42 | 0.53 |
| Sonnet-4 | 0.89 | 0.69 | 0.48 | 0.15 | 0.15 | 0.47 |
| Qwen-VL-Max | 0.54 | 0.65 | 0.40 | 0.95 | 0.64 | 0.64 |
| Grok-4 | 0.43 | 0.41 | 0.36 | 0.97 | 0.61 | 0.56 |
| GLM-4.5V | 0.59 | 0.67 | 0.52 | 0.96 | 0.72 | 0.69 |
| Moonshot | 0.13 | 0.46 | 0.77 | 0.88 | 0.19 | 0.48 |
| *Student Model (7B MLLM)* | | | | | | |
| **Ours** | **0.84** | **0.67** | **0.78** | **0.96** | **0.65** | **0.78** |

Table 2: **Evaluation results of multiple MLLM teachers on classification of MS-CXR-T for comparison.** Top-1 accuracy is applied to evaluate the performance of different methods. The best-performing models are highlighted in red, with the second-best in blue. The comparison MLLMs includes: Claude Sonnet-4 Anthropic (2025), Gemini-2.5 Comanici et al. (2025), GLM-4.5V Team et al. (2025), GPT-5 OpenAI (2025), Qwen-VL-Max Bai et al. (2025), Grok-4 xAI (2025) and Moonshot-v1 AI (2025).

dard MIMIC-CXR. In terms of the edema (Ede.), we argue that excessive disagreement among the teachers is beyond a reconcilable threshold, as indicated in Table 2 with accuracy 0.15 of Sonnet-4 Anthropic (2025) and 0.19 of Moonshot AI (2025) on the edema. It severely hinders the student

model's ability to learn a consistent decision policy. Nonetheless, the distilled student model still outperforms the majority of the teacher models on the edema (Ede.) as illustrated in Table 2.

Beyond comparison with domain-specific methods, we also focus on the robust alignment between drifting MLLM teachers and our distilled student model, as exhibited in Fig.2. The compared MLLM teachers all have huge parameters, while our distilled student is a "small" MLLM with only 7B parameters. Despite it, the distilled student achieves the best average performance with a Top-1 accuracy of 0.78 across all diseases, surpassing every single teacher. It demonstrates that our autonomous distillation empowers the student model to integrate the diverse strengths of multiple teachers, thereby truly learning from all.

Moreover, as shown in Table 2, although the distilled student does not surpass any single teacher MLLM on individual diseases, it consistently achieves the second-best performance across all categories except pleural effusion (PE), underscoring its robustness and stability. Notably, for cases where teacher disagreements are particularly pronounced, such as consolidation (Con.) with an accuracy gap of 0.62 and edema (Ede.) with a gap of 0.57, our approach effectively mitigates unpredictable drifts among MLLM teachers, preventing biased knowledge from infiltrating the student. Even in the extreme case of Pneumonia (Pna.), where only a few positive teachers with accuracy above 0.5 (GPT-5, GLM-4.5V, and Moonshot) are overwhelmed by a larger number of negative teachers, the student is still able to leverage APO-based critique to align more closely with positive feedback despite the prevalence of negative signals.

In addition, the teacher comparison further highlights the substantial drifts among multiple MLLMs, which are manifested not only in the absence of a single teacher that consistently performs best across most diseases, but also in the strikingly large accuracy gaps observed among teachers on individual diseases. These findings underscore the considerable challenges posed by multi-teacher distillation and further validate the central Observation 1.1 of our study.

## 3.2 HARMONIZING CONCEPT ALIGNMENT FOR CONSISTENT THINKING

| | Venue | BLEU-1 | BLEU-2 | BLEU-3 | BLEU-4 | ROUGE-L | METEOR |
|---|---|---|---|---|---|---|---|
| METransformer | CVPR'23 | 0.386 | 0.250 | 0.169 | 0.124 | 0.291 | 0.152 |
| R2GenGPT | MetaRad'23 | 0.408 | 0.256 | 0.174 | 0.125 | 0.285 | 0.167 |
| PromptMRG | AAAI'24 | 0.398 | - | - | 0.112 | 0.268 | 0.157 |
| BtspLLM | AAAI'24 | 0.402 | 0.262 | 0.180 | 0.128 | 0.291 | 0.175 |
| MambaXray | Arxiv'24 | 0.422 | 0.268 | 0.184 | 0.133 | 0.289 | 0.167 |
| **Ours** | **This paper** | **0.463** | **0.272** | **0.210** | **0.152** | **0.298** | **0.213** |

Table 3: **Evaluation results of diagnostic report generation on MIMIC-CXR with various metrics including BLEU-1/-2/-3/-4, ROUGE-L, METEOR and CIDEr.** The best-performing models are highlighted in red. The comparison methods include: METransformer Wang et al. (2023b), R2GenGPT Wang et al. (2023c),BtspLLM Liu et al. (2024) and MambaXray Wang et al. (2024b)

Beyond classification, we further substantiate our core contribution of consistent reasoning, which retains the beneficial CoT across multiple MLLM teachers while suppressing harmful drift. As shown in Table 3, we assess diagnostic report generation on MIMIC-CXR to evaluate the reasoning capabilities of the distilled student model. Evaluation employs BLEU to quantify terminology precision and reasoning coherence, ROUGE-L to assess narrative completeness, and METEOR to capture synonym-aware lexical alignment.

The results demonstrate that our reasoning framework consistently excels across all evaluation metrics, achieving notable gains in BLEU-4 (14.3%), ROUGE-L (2.4%), and METEOR (27.5%), reflecting the consistency and completeness. We attribute this improvement to the "critique" phase within autonomous preference optimization. Unlike conventional knowledge distillation, where teachers primarily provide positive guidance, we reinterpret the drifting outputs of teachers as negative samples, while treating the student's self-distilled, conceptually aligned signals as positive samples for preference learning. This strategy sharpens the decision boundary of the student model, effectively suppresses the transmission of biased information from teachers, and reinforces conceptual consistency within the student.

## 3.3 AMPLIFYING AUTONOMOUS DISTILLATION FOR GENERALIZED REASONING

Furthermore, we assess the generalization ability of our model on downstream tasks through zero-shot multi-label classification across four diverse benchmarks, as reported in Table 4. The results show that our APO-driven MLLMs consistently surpass the second-best baseline, CARZero Lai et al. (2024), across all datasets, highlighting their robustness and strong generalization even when trained under drifting MLLM teachers.

## 3.4 ABLATION STUDIES

Moreover, we conduct ablation experiments on MIMIC-CXR to validate the feasibility and coordination of the multiple teachers (MT) and autonomous preference optimization (APO) under non-stationary knowledge distillation, as presented in Table 5. For the distillation within a single teacher, GPT-5 serves as the teacher due to the best average accuracy among various teachers as exhibited in 2. Moreover, since APO inherently relies on concept alignment across multiple teachers, we do not conduct an ablation study of APO under the setting where MT is absent.

The ablation on MT reveals only marginal overall gains, while performance on most diseases, including Con., PE, Pna. and Pnx. deteriorates. This corroborates our observation that the unpredictable drift among teachers severely disrupts the student's learning and degrades its effectiveness. Besides, compared with MT and SPD, APO delivers significant accuracy gains across all diseases by blocking the transmission of concept drift and enabling the student to constructively learn all teachers. Thus, the consistent, robust, and generalizable improvements confirm that the performance boost arises from APO itself rather than MT.

| Method | Open-I | C'Xray14 | C'Xpert | C'XDet10 |
|---|---|---|---|---|
| BioViL | 0.70 | 0.73 | 0.79 | 0.71 |
| CheXzero | 0.76 | 0.73 | 0.88 | 0.71 |
| MedKLIP | 0.76 | 0.73 | 0.88 | 0.71 |
| KAD | 0.81 | 0.79 | 0.91 | 0.74 |
| CARZero | 0.84 | 0.81 | 0.92 | 0.80 |
| **Ours** | **0.85** | **0.83** | **0.92** | **0.81** |

Table 4: **Evaluation results of zero-shot diseases classification on Open-IDemner-Fushman et al. (2012), ChestXray14 Wang et al. (2017), ChestXpert Irvin et al. (2019) and ChestXDet10 Liu et al. (2020a).** AUC is applied to evaluate the performance of different methods. The best-performing models are highlighted in red. The comparison methods include: BioViL Bannur et al. (2023b), CheXzero Tiu et al. (2022), MedKLIP Wu et al. (2023), KAD Zhang et al. (2023) and CARZero Lai et al. (2024)

| SPD | MT | APO | Con. | PE | Pna. | Pnx. | Ede. | Avg. |
|---|---|---|---|---|---|---|---|---|
| ✓ | - | - | 0.78 | 0.58 | 0.70 | 0.95 | 0.31 | 0.66 |
| ✓ | ✓ | - | 0.77 | 0.49 | 0.69 | 0.94 | 0.51 | 0.68 |
| ✓ | ✓ | ✓ | 0.84 | 0.67 | 0.78 | 0.96 | 0.65 | 0.78 |

Table 5: **Ablation evaluation results on supervised pre-distillation (SPD), multiple teachers (MT) and autonomoous preference (APO) under non-stationary distillation on MIMIC-CXR**. The ✓ denotes that the results are trained with the corresponding module. The results are based on the test split of the MS-CXR-T with the Top-1 accuracy.

## 4 CONCLUSIONS AND LIMITATIONS

In this paper, we introduce autonomous preference optimization (APO), a novel and robust paradigm for generalized knowledge distillation across multiple drifting MLLMs. Grounded in concept drift theory, APO systematically formalizes the biases inherent in the generation of MLLMs and leverages a "learn–compare–critique" paradigm to guide distillation. Through this framework, APO effectively prevents the propagation of concept drift while enabling the student model to learn from all teachers, assimilating the complementary strengths in a constructive manner.

We envision that this work will stimulate further progress in knowledge distillation for MLLMs, particularly in addressing domain-specific biases. Looking ahead, our future efforts will concentrate on enhancing the efficiency and reducing the computational cost of distillation in large-scale multimodal settings.

ETHICS STATEMENT

This work adheres to the ICLR Code of Ethics. In this study, no human subjects or animal experimentation was involved. All datasets used, including CXR-MAX, MIMIC-CXR, MS-CXR-T, Open-I, ChestXray14, ChestXpert and ChestXDet10, were sourced in compliance with relevant usage guidelines, ensuring no violation of privacy. We have taken care to avoid any biases or discriminatory outcomes in our research process. No personally identifiable information was used, and no experiments were conducted that could raise privacy or security concerns. We are committed to maintaining transparency and integrity throughout the research process.

REPRODUCIBILITY STATEMENT

We have made every effort to ensure that the results presented in this paper are reproducible. All code and datasets have been made publicly available in an anonymous repository to facilitate replication and verification. The experimental setup, including training steps, model configurations, and hardware details, is described in detail in the paper. We have also provided a full description of autonomous distillation to assist others in reproducing our experiments.

Additionally, our contributed dataset, CXR-MAX, is publicly available, ensuring consistent and reproducible evaluation results.

We believe these measures will enable other researchers to reproduce our work and further advance the field.

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

# A    RELATED WORKS

## A.1    CONCEPT DRIFT

Building on an extensive body of work, Lu et al. Lu et al. (2019; 2020) provide a systematic survey that organizes concept drift mitigation into three dominant families: error rate–driven adaptation Wang et al. (2024a); Jiao et al. (2024), distribution-aware approaches Yang et al. (2025a); Cerqueira et al. (2023); Yang et al. (2023), and multi-hypothesis frameworks Yu et al. (2024; 2022). Our study is situated within the distribution-oriented stream, which is notable for coupling rigorous statistical tests with broad representational power, thereby enabling not only accurate detection of drift but also its nuanced characterization along temporal, spatial, and quantitative axes. By supporting fine-grained diagnostics such as the timing of drift onset, the attribution of drift to specific feature subspaces, and the assessment of its magnitude, distribution-based methods provide a principled foundation for adaptive systems that demand both interpretability and precise recalibration in the presence of evolving data.

Ongoing research on concept drift adaptation has produced a wide spectrum of refined techniques designed for increasingly complex learning environments. Among them, the Online Boosting Adaptive Learning (OBAL) framework Yu et al. (2024) offers a two-stage pipeline for multistream classification, beginning with Adaptive Covariate Shift Adaptation (AdaCOSA) to capture evolving inter-stream correlations, and subsequently employing a Gaussian Mixture Model–driven weighting scheme to counter asynchronous distributional changes. In the multimodal landscape, CDMLLM Yang et al. (2025a) highlights the susceptibility of vision–language models to drift-induced biases that arise during both pre-training and fine-tuning, and proposes a unified remedy that integrates T-distribution calibration for long-tailed scenarios with explicit out-of-distribution detection, thereby reinforcing alignment stability. Beyond single-stream settings, GDDM Yu et al. (2023) contributes a distribution-free statistical mechanism for uncovering subtle group-level shifts in multi-stream data, relying on adaptive hypothesis testing to achieve robust detection. Anticipatory strategies have also been explored, most notably in DDG-DA Li et al. (2022-06-28), which projects potential environmental evolution by coupling predictive factor analysis with synthetic data generation, creating a principled bridge between current observations and future distributional states. Complementing these supervised paradigms, STUDD Cerqueira et al. (2023) introduces an unsupervised teacher–student discrepancy model that measures predictive consistency to flag drift without dependence on annotated labels, thereby reconciling sensitivity to distributional change with the practical limitations of real-world deployment.

## A.2    KNOWLEDGE DISTILLATION FOR LLMS AND MLLMS

Knowledge distillation (KD) Hinton et al. (2015) has become a central paradigm for transferring knowledge from large teachers to compact students. In NLP, KD was initially used to compress BERT-like models Sun et al. (2019); Jiao et al. (2020); Sanh et al. (2019), enabling efficiency without major performance loss. More recently, KD has been extended to large language models (LLMs), where the focus is not only on compression but also on improving reasoning robustness, alignment, and sample efficiency. For example, self-distillation and on-policy KD Agarwal et al. (2023); Yang et al. (2024c) help students learn from their own generated trajectories or mitigate distribution gaps during fine-tuning. Other works integrate distillation with instruction tuning Wang et al. (2023a); Zhou et al. (2023), preference optimization Ouyang et al. (2022); Bai et al. (2022), or symbolic reasoning West et al. (2022); Gu et al. (2023), highlighting the versatility of KD for enhancing LLM performance.

In multi-modal contexts, KD plays an essential role in bridging cross-modal representations. Vision-language models such as CLIP Radford et al. (2021) and FILIP Yao et al. (2022) motivated distillation strategies for multi-modal grounding. Frameworks like BLIP-2 Li et al. (2023a), LLaVA Liu et al. (2023), InstructBLIP Dai et al. (2023), and LLaVA-MoD Shu et al. (2024) employ KD from strong encoders or larger MLLMs to align modalities, compress architectures, or improve downstream reasoning. Recent innovations include Align-KD Feng et al. (2025a) and MoVE-KD Cao et al. (2025b), which distill cross-modal alignment knowledge and ensemble signals from multiple visual encoders, respectively, demonstrating the growing interest in efficient and robust MLLM distillation. KD has also been explored for domain-specific applications, such as robotic surgery VQA Chen et al. (2024b) and medical multimodal models Shen et al. (2025b).

A complementary direction is multi-teacher distillation, which synthesizes complementary strengths from multiple teachers. Early studies in CV You et al. (2017); Son et al. (2021); Yuan et al. (2021); Liu et al. (2020b); You et al. (2018) inspired recent extensions to LLMs and MLLMs. For example, Gu et al. Gu et al. (2025b) propose multi-MLLM distillation for out-of-context news detection, while multi-teacher continual learning frameworks Chen et al. (2024b) address streaming data in specialized domains. Moreover, recent benchmarks on distillation under distribution shift Zhang et al. (2025a) highlight the challenge of biased or drifting teacher supervision, which has not been systematically solved in multi-modal KD.

Overall, KD has evolved from compression to a general tool for alignment, reasoning transfer, and cross-modal adaptation. Yet, most methods remain constrained by single-teacher assumptions or overlook distribution drift across modalities. These gaps motivate our work on robust multi-teacher distillation for MLLMs, explicitly addressing the challenges of bias inheritance and teacher drift.

### A.3 REINFORCED FINE-TUNING IN LLMs

The role of reinforcement learning (RL) in shaping post-training alignment of large language models (LLMs) has advanced significantly since OpenAI's pioneering work on Reinforcement Learning from Human Feedback (RLHF) Christiano et al. (2017), which introduced a paradigm for aligning model behavior with human values Ouyang et al. (2022). Initial implementations, such as OpenAI-o1 Jaech et al. (2024), demonstrated the practical utility of preference-driven modeling, yet the reliance on large-scale human annotation quickly revealed severe limitations in cost and scalability. These constraints have spurred a transition toward automated reward construction using pre-trained systems, opening the door to a new generation of alignment methods. Bai et al.'s Bai et al. (2022) constitutional framework, for example, relies on sparse natural language feedback as an indirect supervisory signal, while DeepSeek's research line illustrates a staged trajectory: beginning with a purely RL-based baseline (R0), and subsequently extending to the R1 system Guo et al. (2025), which cycles between supervised fine-tuning and their GRPO optimization scheme Shao et al. (2024). This cyclic design improved generalization capacity and marks a broader trend toward increasingly autonomous alignment pipelines that minimize human involvement while retaining robust performance.

Concurrently, alignment research has diversified through a range of novel paradigms that extend beyond the classical RLHF formulation. ReST Gulcehre et al. (2023) advances iterative self-training by generating policy-driven samples and refining them via offline RL, while DPO Rafailov et al. (2023b) reconceptualizes the task as direct optimization of preferences through implicit reward modeling. Complementary efforts include Rejection Sampling Fine-Tuning (RSFT) Yuan et al. (2024), which augments supervised training with carefully filtered reasoning trajectories, and ReFT Trung et al. (2024), which couples supervised fine-tuning initialization with PPO-based exploration to progressively expand reasoning capabilities. Extending these principles to multimodal contexts, Visual-RFT Liu et al. (2025) adapts GRPO-driven strategies for visual-language alignment under limited data regimes, whereas B-STaR Zeng et al. (2025) introduces dynamic configuration mechanisms that balance exploration and exploitation for self-improving systems. Methodological innovation has also been paralleled by advances in evaluation: Qwen-Math-PRM Zhang et al. (2025b) integrates Monte Carlo estimation with LLM-as-judge consensus, building a hierarchical framework that captures both stepwise reasoning fidelity and holistic solution quality. Along a similar line, ViLaM Yang et al. (2024b) leverages reinforcement learning for unsupervised visual grounding, tackling the challenges of open-world multimodal understanding.

## B CXR-MAX DATASET

In this section, we showcase the samples utilized for training and validation in our study, generated by various MLLMs, with the image and ground truth of the radiology report. And the prompt we used to various MLLMs is:

*""This is a patient's chest DR image. The patient has been diagnosed with ¡diseases¿. Can you find the basis for the diagnosis in the image?"*

**Radiology Report**

Findings: There is unchanged moderate left lower lobe atelectasis and bilateral pleural effusions. Linear atelectases in the left mid lung field are again seen, essentially unchanged. Lesion in the right seventh rib is unchanged. Pleurx catheter is seen, unchanged in position at the base of the left lung. There is no evidence of pneumothorax. Pacer is seen with leads appropriately placed within the right and left atria. Mediastinal silhouette is within normal limits with a calcified aorta.

**Claude Sonnet-4**

Looking at this chest X-ray, I can identify several findings that support the diagnoses of atelectasis and pleural effusion:

**Evidence of Atelectasis:**

1. **Volume loss in the right hemithorax** - The right side appears smaller compared to the left

2. **Mediastinal shift** - The mediastinum (including the trachea and heart) appears shifted toward the right side, which is characteristic of right-sided volume loss

3. **Elevated right hemidiaphragm** - The right diaphragm appears higher than the left

4. **Increased opacity in the right lung** - Areas of collapsed lung tissue appear more dense/white

**Evidence of Pleural Effusion:**

1. **Blunting of the right costophrenic angle** - The normally sharp angle where the diaphragm meets the chest wall is obscured

2. **Increased density at the right lung base** - There appears to be fluid accumulation in the lower right pleural space

3. **Meniscus sign** - The fluid appears to have the characteristic curved upper border typical of pleural effusion

The combination of these findings - volume loss with mediastinal shift (atelectasis) and basal opacity with costophrenic angle blunting (pleural effusion) - provides clear radiographic evidence supporting both diagnoses. The presence of what appears to be a pacemaker or similar device in the right chest area is also noted.

**GPT-5-nano**

Key findings that support atelectasis and pleural effusion:

- Left subclavian pacemaker/ICD in place (device visible in the left chest).

- Basilar opacities at the left lower lung suggesting subsegmental/plate-like atelectasis with some volume loss (diaphragm slightly elevated on the left).

- Blunting of the left costophrenic angle with posterior layering of fluid consistent with a small left pleural effusion (meniscus sign may be subtle on this view).

Right lung and mediastinal contours are otherwise relatively unremarkable.

In short: left basilar atelectasis with a small left pleural effusion. Correlate clinically; consider follow-up radiographs or ultrasound/CT if the clinical picture warrants.

### GLM-4.5V

To determine the basis for diagnosing **Atelectasis** and **Pleural Effusion** in the chest X - ray, we analyze key imaging features:

**1. Pleural Effusion:**

Pleural effusion is identified by **blunting of the costophrenic angles** (the normally sharp, downward - pointing angles where the diaphragm meets the rib cage). In this image, the right costophrenic angle appears blurred and loses its typical sharp contour, which is a classic sign of fluid accumulation in the pleural space.

**2. Atelectasis:**

Atelectasis refers to partial lung collapse, which shows as **increased opacity (whiteness)** in the affected lung regions due to reduced air content. In this X - ray, there are areas of increased density in the lower lung fields (particularly on the right side), indicating alveolar collapse and loss of normal lung aeration.

In summary, the diagnostic clues are the **blunted right costophrenic angle (pleural effusion)** and **increased opacification in the lower lung zones (atelectasis)**.

The answer is The basis for the diagnosis includes blunted right costophrenic angle (indicating pleural effusion) and increased opacification in the lower lung zones (indicating atelectasis) .

### Gemini-2.5-flash-lite

Based on the chest DR image, here are the findings that support the diagnoses of Atelectasis and Pleural Effusion:

**1. Pleural Effusion:**

- **Right Costophrenic Angle Blunting:** The right costophrenic angle is completely blunted and obscured by fluid.

- **Meniscus Sign:** There is a dense, homogeneous opacity at the right lung base with a characteristic curvilinear upper border (meniscus sign) that tracks upwards along the lateral chest wall. This indicates free fluid in the pleural space.

- **Obscured Diaphragm**: The right hemidiaphragm is completely obscured by the effusion.

This combination clearly indicates a significant right-sided pleural effusion.

**2. Atelectasis:**

- **Volume Loss in Right Hemithorax:** #blank

- **Mediastinal Shift:** The mediastinum, including the cardiac silhouette and potentially the trachea (though less clearly seen), is shifted towards the right (the affected side). This indicates a reduction in lung volume on the right.

- **Reduced Right Lung Volume:** The overall right lung field appears smaller and more collapsed compared to the left lung.

- **Increased Opacity/Consolidation:** The remaining lung parenchyma in the right hemithorax, particularly superior to the pleural effusion, appears denser and more opaque than normal lung, consistent with collapsed or consolidated lung tissue.

The atelectasis is likely, at least in part, compressive due to the large pleural effusion, but could also have an obstructive component.

In summary, the image demonstrates a large right-sided pleural effusion, causing blunting of the costophrenic angle and a meniscus sign, accompanied by signs of volume loss in the right hemithorax (mediastinal shift to the right) and increased density of the remaining lung, all consistent with a diagnosis of right-sided pleural effusion and atelectasis.

### Moonshot-v1

As an AI, I'm not capable of providing medical diagnoses or interpreting medical images with the same level of expertise as a trained medical professional. However, I can provide general information on what to look for in a chest X-ray when atelectasis and pleural effusion are suspected.

Atelectasis is a condition where one or both lungs partially or completely collapse. On an X-ray, this might appear as an area of increased opacity (darker area) in the lung field, which could indicate that the lung tissue is not fully expanded.

Pleural effusion refers to the buildup of excess fluid in the pleural cavity, the space between the lungs and the chest wall. This can cause the lung to partially collapse and may appear on an X-ray as a layering density at the lung base or costophrenic angle, which could be seen as a meniscus sign (a curved line of increased density).

It's important to note that accurate diagnosis and interpretation of medical images should be done by a qualified radiologist or physician who can consider the full clinical context and other diagnostic information. If you have concerns about a medical condition, please consult a healthcare professional.

---

**Qwen-VL-Max**

The chest X-ray (DR image) shows several findings consistent with **atelectasis** and **pleural effusion**, particularly on the **left side**. Here's the basis for each diagnosis:

**1. Pleural Effusion (Left Side)**

**Key Findings:**

- **Blunting of the left costophrenic angle:** The sharp angle between the diaphragm and rib cage on the left is obscured or rounded, which is a classic sign of pleural fluid accumulation.

- **Homogeneous opacity in the lower left hemithorax:** There is increased density in the left lower lung field, especially near the base, which does not follow the bronchovascular markings. This suggests fluid rather than lung parenchymal disease.

- **Meniscus sign:** A concave upper border of the fluid collection (though less clearly visible here due to overlying structures), often seen in moderate pleural effusions.

- **Elevation of the left hemidiaphragm:** Fluid can push the diaphragm upward.

Note: The presence of a **cardiac pacemaker** on the left side may partially obscure the view, but the effusion is still evident.

**2. Atelectasis (Left Lower Lobe Likely)**

**Key Findings:**

- **Volume loss in the left lower lung:** The left hemithorax appears compressed, and there is a shift of mediastinal structures slightly toward the left (e.g., trachea and heart), indicating volume loss.

- **Loss of normal lung markings in the affected area:** The lung tissue appears denser and consolidated, with reduced air content.

- **Possible displacement of the minor fissure**: (if visible): In lobar atelectasis, the fissures may be displaced due to collapse.

The atelectasis likely results from **compression by the pleural effusion** —this is known as **compressive atelectasis**.

**Additional Notes:)**

- The **right lung** appears relatively clear, though some mild interstitial markings may be present, possibly due to chronic changes or early involvement.

- The **heart silhouette** appears enlarged, which could be due to the effusion pushing the heart laterally, or it may reflect underlying cardiomegaly.

- The **pacemaker device** is well-positioned, with leads extending into the right atrium and ventricle; no evidence of lead dislodgement.

**Summary:)**

✅ **Pleural effusion**: Left-sided, based on blunted costophrenic angle, homogeneous opacity, and diaphragmatic elevation.

✅ **Atelectasis**: Likely left lower lobe, secondary to compressive atelectasis from the adjacent pleural effusion. These findings are typical of conditions such as pneumonia, heart failure, malignancy, or post-operative states, depending on clinical context.

**Recommendation**: Further evaluation with ultrasound or CT thorax may help confirm the nature and extent of the effusion and guide management (e.g., thoracentesis).

> **Grok-4**
>
> I can explain the basis for the diagnosis of atelectasis and pleural effusion based on an analysis of the chest X-ray (CXR) you provided. I'll keep this concise, objective, and focused on the visible findings in the image that support the diagnosis. Please note that I am not a substitute for the patient's actual treating physician, and CXR interpretation should always be correlated with the full clinical picture (e.g., symptoms, physical exam, labs, and possibly additional imaging like a CT scan or ultrasound for confirmation). If this is your CXR or that of someone you know, please discuss it directly with a doctor for personalized advice.
>
> **Key Observations in the Image:**
>
> This is a frontal (posteroanterior or PA) view of the chest. The lungs, heart, mediastinum (central chest structures), diaphragm, and bony thorax are visible. There's also an incidental implanted cardiac device (e.g., pacemaker or defibrillator) on the left side with leads extending into the heart—this is unrelated to the diagnosis but worth noting as it can sometimes complicate interpretation.
>
> **1. Basis for Pleural Effusion:**
>
> **Location**: Primarily on the right side (which appears on the left side of the image, as CXRs are conventionally viewed as if you're facing the patient).
>
> **Key Findings**:
>
> - Increased opacity (whitish haziness) at the right lung base, obscuring the normal sharp outline of the right hemidiaphragm (the dome-shaped muscle separating the chest from the abdomen).
>
> - Blunting or filling-in of the right costophrenic angle (the sharp angle where the diaphragm meets the chest wall). Normally, this angle is very sharp and black (air-filled); here, it's filled with a denser, hazy material, which is classic for fluid accumulation.
>
> - A subtle "meniscus sign" (curved upper border of the opacity) at the right base, suggesting free-flowing fluid layering dependently due to gravity in this upright view.
>
> - The opacity is homogenous (evenly dense) and respects the lung boundaries without air bronchograms (air-filled airways standing out against the density), which fits with fluid in the pleural space rather than within the lung tissue itself.
>
> **Why This Indicates Pleural Effusion**: Pleural effusion is an abnormal collection of fluid between the lung and chest wall. In an upright CXR, it typically settles at the base of the lung, causing these exact changes. The amount here appears moderate (not massive, as the mediastinum isn't shifted much), but it's enough to cause visible changes. Common causes include infection, heart failure, malignancy, or inflammation, but the CXR alone doesn't specify the cause.
>
> **2. Basis for Atelectasis:**
>
> **Location**: Also primarily involving the right lower lobe (base of the right lung).
>
> **Key Findings**:
>
> - Loss of volume and increased density in the right lower lung zone, making it appear smaller and whiter compared to the left lung (which is more uniformly black and air-filled).
>
> - Crowding or absence of normal lung markings (vascular structures) in the affected area, with the remaining lung tissue appearing compressed.
>
> - Subtle elevation or silhouetting of the right hemidiaphragm, suggesting collapse of the adjacent lung tissue.
>
> - No obvious shift of the trachea or mediastinum (which would indicate more severe collapse), but the changes are localized to the base.
>
> **Why This Indicates Atelectasis**: Atelectasis is partial or complete collapse of lung tissue, leading to reduced air content and increased density on CXR. It often occurs adjacent to a pleural effusion (the fluid compresses the lung), which is likely the case here—the effusion is probably contributing to compressive atelectasis. Other signs like air bronchograms are not prominent, suggesting it's not primarily consolidative (e.g., from pneumonia).
>
> **3. Other Supporting Details:**
>
> - **Bilateral Comparison**: The left lung appears relatively normal (darker and more aerated), which highlights the asymmetry and makes the right-sided abnormalities more evident.
>
> - **No Contradictory Findings**: There are no signs of other issues (e.g., pneumothorax, large masses, or fractures) that might mimic these changes.
>
> - **Technical Quality**: The image is well-exposed and positioned, reducing the chance of artifacts mimicking disease.

## C  IMPLEMENTATION DETAILS

In this section, implementation details are provided.

In terms of the supervised fine-tuning progress, the hyperparameters are presented in Table 6. Qwen2.5-VL (7B) Bai et al. (2025) is applied as our pre-trained model. During the SPD, we utilize the AdamW optimizer, which is configured with a cosine annealing schedule as the learning policy. The initial learning rate is set to $1 \times 10^{-4}$, and the AdamW optimizer is employed with hyperparameters $\beta = (0.9, 0.98)$. Additionally, we set the weight decay to 0.05 and the dropout rate to 0.1. During the first 20 warm-up steps, the learning rate increases to $1 \times 10^{-4}$, and subsequently decays to $10^{-7}$. Unless otherwise specified, the supervised pre-distillation of our multi-modal large language model consists of 10,686 steps, executed on $2 \times 2$ NVIDIA A100 GPUs.

Table 6: The training hyperparameters of our MLLM.

| Supervised Pre-Distillation | | Autonomous Preference Optimzation | |
| --- | --- | --- | --- |
| Training Steps | 10,686 | Training Steps | 12,132 |
| Warmup Steps | 20 | Warmup Steps | 0 |
| Warmup Ratio | 0.05 | Optimizer | AdamW |
| Optimizer | AdamW | Learning Rate | 2e-5 |
| Learning Rate | 1e-4 | Learning Rate Decay | Cosine |
| Learning Rate Decay | Cosine | Adam $\beta$ | (0.9, 0.98) |
| Adam $\beta$ | (0.9, 0.98) | Weight Decay | 0.05 |
| Weight Decay | 0.05 | Batch Size | 2 |
| Batch Size | 2 | | |

While in the autonomous preference optimization (APO), the initial learning rate is reduced to $2 \times 10^{-5}$ without the warmup, with the batch size of 2. The visual encoder and text decoder are frozen out of the training. The reinforced custom-tuning consists of 12,132 steps, executed on $2 \times 2$ NVIDIA A100 GPUs. Other training parameters are the same as the fine-tuning.

## D  MORE RESULTS

| | Venue | BLEU-1 | BLEU-2 | BLEU-3 | BLEU-4 | ROUGE-L | METEOR |
| --- | --- | --- | --- | --- | --- | --- | --- |
| METransformer | CVPR'23 | 0.386 | 0.250 | 0.169 | 0.124 | 0.291 | 0.152 |
| R2GenGPT | MetaRad'23 | 0.408 | 0.256 | 0.174 | 0.125 | 0.285 | 0.167 |
| BtspLLM | AAAI'24 | 0.402 | 0.262 | 0.180 | 0.128 | 0.291 | 0.175 |
| MambaXray | Arxiv'24 | 0.422 | 0.268 | 0.184 | 0.133 | 0.289 | 0.167 |
| CounterfactRFT | NeurIPS'25 | 0.426 | 0.288 | 0.186 | 0.155 | 0.421 | 0.286 |
| **Ours** | **This paper** | **0.563** | **0.372** | **0.270** | **0.192** | **0.298** | **0.213** |

Table 7: **Evaluation results of diagnostic report generation on MIMIC-CXR with various metrics including BLEU-1/-2/-3/-4, ROUGE-L, METEOR and CIDEr.** The best-performing models are highlighted in red. The comparison methods include: R2Gen Chen et al. (2020), PPKED Liu et al. (2021b), AlignTrans You et al. (2021), CMCL Liu et al. (2021a), Clinical-BERT Yan & Pei (2022) ,METransformer Wang et al. (2023b) ,DCL Li et al. (2023b), R2GenGPT Wang et al. (2023c) ,PromptMRG Jin et al. (2024) ,BtspLLM Liu et al. (2024) ,MambaXray Wang et al. (2024b), CounterfactRFT Yang et al. (2025b)

## E  LLM USAGE

Large Language Models (LLMs) were used to aid in the writing and polishing of the manuscript. Specifically, we used an LLM to assist in refining the language, improving readability, and ensuring clarity in various sections of the paper. The model helped with tasks such as sentence rephrasing, grammar checking, and enhancing the overall flow of the text.

It is important to note that the LLM was not involved in the ideation, research methodology, or experimental design. All research concepts, ideas, and analyses were developed and conducted by

the authors. The contributions of the LLM were solely focused on improving the linguistic quality of the paper, with no involvement in the scientific content or data analysis.

The authors take full responsibility for the content of the manuscript, including any text generated or polished by the LLM. We have ensured that the LLM-generated text adheres to ethical guidelines and does not contribute to plagiarism or scientific misconduct.

