# OpenReview forum: "Learning from All: Concept Alignment for Autonomous Distillation from Multiple Drifting MLLMs"
_ICLR.cc/2026/Conference — Submitted to ICLR 2026_

### Official Review · Reviewer_XwXM · 2025-10-23

**Soundness:** 1
**Presentation:** 1
**Contribution:** 1
**Rating:** 2
**Confidence:** 3

**Summary:**

This paper introduces APO, a new framework for distilling reasoning capabilities from multiple MLLMs that exhibit conceptual drift, defined as variability in their reasoning behaviors or conclusions. The core idea is that APO aggregates all available reasoning trajectories and learns to prefer the self-distillation as positive signals against all negative trajectories. This approach treats distillation as a preference optimization problem, aligning the student model’s reasoning trajectory with the highest-quality outputs among drifting teachers, in a “learn-compare-critique” paradigm. The method is tested on a newly constructed dataset, CXR-MAX, based on chest X-ray interpretation, and shows improvements in accuracy.

**Strengths:**

- Conceptually interesting framework that reframes multiple teacher KD as a PO problem over reasoning trajectories and models a *drifting* behavior, although this drifting is not well elaborated.
- The paper provides a clear experimental setup, with the introduction of CSR-MAX dataset to evaluate alignment in medical reasoning.

**Weaknesses:**

There are several weaknesses regarding the lack of clarity in methods, the justification on the problem setting, and the rationale about the limited application of the proposed method.

- The paper is using these kind of terms: “drifting teachers”, “drifting MLLMs”, “inter-model drift”, “concept drift”, but the central concept of “drift” is not well-defined or formalized. A formal mathematical definition of drifting or empirical characterization of what and how constitutes “drift” would help.
- Example in Figure 1 (b) just looks like a standard error or hallucination commonly observed in LLMs. It fails to provide evidence of concept drift, or that this error came from the drifting teachers.
- The paper assumes a multi-teacher distillation setup which is a very specific framework. It is unclear why this setup is necessary when multiple MLLM teachers exhibit inter-model drift. In practice, single-teacher KD is used in general, and the proposed scenario of drifting across many MLLMs is not well justified.
- The authors claim that distilled students exhibit concept drift after learning from misaligned teachers, yet APO selects the student’s own trajectory as the preferred one during the optimization. If the student is already drifting, why is its reasoning preferred?

Experimental demonstration is only limited to chest X-ray image domain, which is very specific and narrow.

- Since APO is presented as a general framework for MLLM distillation, experiments on broader tasks (e.g., general image classification, natural language reasoning, or multi-step math and coding problems) would provide more compelling evidence of its utility. I cannot find the reason why the authors have applied it exclusively on the medical domain.
- No quantitative evaluation is provided on the degree of concept drift or how much APO reduces this drift. Some metric to evaluate teacher-teacher drifting and teacher-student alignment over time or across tasks would strengthen the claims.


### Writing and Attribution Concerns

There are notable similarities in structure between this paper and prior work, particularly Yang et al. (2025b), which has been cited in this paper. [Walking the Tightrope: Disentangling Beneficial and Detrimental Drifts in Non-Stationary Custom-Tuning, Yang et al., 2025]

For example, the first contribution point in Introduction says:

> We establish a novel theoretical framework that casts the autoregressive generation of reasoning trajectories in MLLMs within the perspective of concept drift theory, providing a principled foundation for understanding and analyzing knowledge distillation from multiple drifting teachers.

**Yang et al. (2025b)**:

> we establish a novel theoretical framework that formalizes autoregressive token generation in MLLMs through the lens of concept drift theory, enabling systematic identification and causal analysis of detrimental reasoning divergence during non-stationary reinforced custom-tuning.

The last contribution point in Introduction says:

> As a pioneer contribution to the community, we introduce CXR-MAX (Multi-teahers Alignment X-rays), a large-scale dataset comprising 170,982 distillation results of reasoning trajectories derived from publicly accessible MLLMs based on MIMIC-CXR.

**Yang et al. (2025b)**:

> As a pioneer contribution to the community, we introduce CXR-CounterFact (CCF), a large-scale dataset comprising 320,416 meticulously curated counterfactual reasoning trajectories derived from MIMIC-CXR.

While the details of methodology differ, the overall framing, structure, and phrasing appear heavily relied on the previous work, raising concerns about originality in presentation. Many parts of Abstract, Introduction, and main figures look too similar with a few technical modifications.

**Questions:**

See weaknesses

---

### Official Review · Reviewer_GPmW · 2025-10-30

**Soundness:** 2
**Presentation:** 2
**Contribution:** 2
**Rating:** 4
**Confidence:** 2

**Summary:**

This paper discusses the concept drift problem in knowledge distillation of multimodal large language models (MLLM). Through the analysis of the connection between concept drift and knowledge distillation, the authors introduce the “learn–compare–critique” paradigm to tackle the issue. The resulting method, autonomous preference optimization (APO), trains the student with self relection over the drifting inference for concept alignment. Experiments demonstrate the effectiveness of APO on knowledge distillation tasks. The authors also contribute to a large-scale dataset called CXR-MAX.

**Strengths:**

1. The paper is well structured
2. Extensive experiments demonstrate the effectiveness of the proposed method.

**Weaknesses:**

1. **Unclear notation**: Definition 2.1 and 2.2 are not clear to me. In particular, the definition of $P_{i, j}$ and $P_i$ are not explained anywhere, do they refer to the joint probability distribution across teachers? More broadly, the jump from Eq. (1) to Definition 2.1 and then to multi-stream drift in Definition 2.2 is not intuitive.

2. **Unclear connection between concept drift and MLLM**: the problem of concept drift might occur naturally in multi-agent distillation. And it seems that there is not special connection to "multimodality"

3. The proposed algorithm in Section 2.3 is a natural extension of DPO, which seems not relevant to the definition in section 2.1

4. The figures in the paper contain a lot of annotations, but a more clear pipeline would help readers' understanding.

5. The experiments are entirely focused on chest X-rays.

**Questions:**

1. How does $t^{+}$ generated exactly in Eq (7)? Does it aggregate the results of $\mathcal{T}$? If it's the response from $\pi_{st} $, given input image $v$ and prompt $l$, then it should be $t^+ \sim \pi_{st} (\cdot|v,l)$.

2. Experiments are mainly done on the domain of Chest X-ray. What would the performance of APO be on other general domains of visual reasoning?

3. Is APO sensitive to the choice of teacher models?

---

### Official Review · Reviewer_L18q · 2025-11-01

**Soundness:** 3
**Presentation:** 3
**Contribution:** 3
**Rating:** 6
**Confidence:** 3

**Summary:**

The paper studies the problem of knowledge distillation from multiple multimodal large language models (MLLMs). The authors observe that the reasoning trajectories of different teacher models can change inconsistently across models or over time, and that such concept drift can propagate to student models during distillation. To address this issue, the paper proposes a “learn–compare–critique” pipeline. The student model first learns from multiple MLLM teachers; then it performs self-distillation to align and identify inconsistent teacher outputs. Finally, through a preference optimization step, the student reinforces alignment with stable reasoning outputs while down-weighting drifted or biased outputs.

For experiments, the authors construct the CXR-MAX dataset, which is an extension of the MIMIC-CXR dataset by adding reasoning trajectories about clinical chest X-ray interpretation from multiple MLLM teachers. Results show that the proposed method outperforms other existing distillation methods, while achieving performance comparable to or exceeding that of the original teacher models.

**Strengths:**

1. The paper studies an interesting and practical problem of distilling knowledge from multiple MLLMs, and the identified challenge of concept drift across models or over time is reasonable and well motivated.

2. The proposed “learn–compare–critique” pipeline is conceptually sound and clearly presented.

3. The experimental results are promising, showing that the proposed approach outperforms both previous distillation methods and the original teacher MLLMs. The ablation study provides useful evidence on the contribution of each component and helps analyze the effectiveness of the proposed framework.

**Weaknesses:**

1. All experiments are conducted in the medical imaging domain, specifically chest X-ray reasoning. The paper does not provide evidence that the proposed approach generalizes to other modalities (e.g., vision–language reasoning, VQA) or non-medical tasks, which limits its demonstrated scope.

2. The study uses only one base model (Qwen2.5-VL, 7B) as the student backbone throughout experiments. It remains uncertain whether the proposed framework would behave similarly to other models or model scales.

3. The paper’s citation style does not follow standard formatting, and some theoretical sections (e.g., formal definitions of concept drift) are expressed in dense, overly complex language. Simplifying these formulations and improving readability would make the paper more accessible.

4. The paper could be strengthened by discussing the method's limitations.

**Questions:**

How sensitive is APO to the number or quality of teacher models? Would it still help if one teacher is significantly weaker?

---

### Official Review · Reviewer_ppCm · 2025-11-02

**Soundness:** 2
**Presentation:** 3
**Contribution:** 2
**Rating:** 2
**Confidence:** 3

**Summary:**

- This paper aims to address the challenge of knowledge distillation from multiple, heterogeneous MLLMs. The main challenge is the concept drift problem, where the teacher models provide conflicting information that can confuse the student model.
- To tackle this, this work proposes a novel three-stage "learn-compare-critique" paradigm called Autonomous Preference Optimization (APO). The student model first learn a broad knowledge via standard supervised distillation from all teachers. Second, it compares and aggregates the teachers' outputs and performs self-distillation to generate a unified reasoning trajectory. Finally, it critiques the initial knowledge by using the consensus trajectory as a preferred sample and the individual teacher outputs as negative samples using a simple contrastive learning loss.

**Strengths:**

- The paper is well-written and easy to understand. The concept drift problem is clear and easy to follow.
- The core idea is straightforward: using the student policy model to rephrase the distilled multiple teacher model responses and then learning from it, whether using imitation learning (SFT) or using offline RL (DPO), which is used in this work.

**Weaknesses:**

- How the model actually creates the consensus answer from all the different teacher outputs. Is the text from all teachers combined?  If so, how to do this when the multiple teacher outputs are too long for the student model?
- The self-distillation process is a critical step in this work. However, the student model does not undergo specific training on the aggregation task of multiple teacher outputs; how can it generate positive samples for subsequent DPO training?
- The APO framework treats the outputs from all strong teacher models (e.g., GPT-5) as negative samples, which assumes that the pre-distilled small student model can provide better responses based on the aggregation outputs. This claim should have a solid experiment to verify (e.g., using a rule/model-based score to justify this).
- This work lacks a lot of baselines to be compared with in the main results: (1) single teacher distillation (KL-based / response-based); (2) naive multi-teacher distillation (KL-based / response-based); (3) distill-then-merge methods.

**Questions:**

Same as weakness.

---

### Official Review · Reviewer_5ziE · 2025-11-03

**Soundness:** 2
**Presentation:** 1
**Contribution:** 2
**Rating:** 2
**Confidence:** 4

**Summary:**

This paper addresses the underexplored problem of knowledge distillation from multiple drifting MLLMs, where inconsistent reasoning trajectories across teachers cause concept drift and bias propagation. The authors propose APO, a “learn–compare–critique” paradigm that enables the student model to self-distill and align reasoning concepts autonomously. Experiments show that this method has certain effectiveness.

**Strengths:**

1. The paper establishes a clear theoretical link between concept drift and knowledge distillation, extending the KD paradigm to multi-teacher and non-stationary environments.
2. The introduction of the CXR-MAX dataset provides a new and valuable benchmark for the medical imaging community.

**Weaknesses:**

1. The paper’s motivation is unclear. Although the authors claim to study the distillation of MLLMs, they do not compare the method with other mainstream methods[1,2,3,4,5,6] in the distillation community. In my view, this work reads more like a medical image understanding paper rather than a genuine distillation study.
2. All experiments are conducted solely on the CXR-MAX dataset, making it difficult to demonstrate the general effectiveness of the proposed method.
3. The assumptions are overly strong. In Equation (4), the authors assume that the inference trajectories of teacher models are independent. This assumption is unrealistic in practice since different MLLMs often originate from similar pretraining distributions.
4. The paper claims theoretical contributions, but the proposed APO objective is mainly a reformulation of DPO and lacks a theoretical analysis of convergence and stability under multi-teacher drift scenarios.
5. The notation in the theoretical section is confusing, with many symbols left undefined or unexplained, making the derivations hard to follow.
6. The font size in Figure 2 is too small, making it difficult to read.

[1] f-Divergence Minimization for Sequence-Level Knowledge Distillation. ACL 2023.

[2] DistiLLM: Towards Streamlined Distillation for Large Language Models. ICML 2024.

[3] MiniLLM: Knowledge Distillation of Large Language Models. ICLR 2024.

[4] Rethinking Kullback-Leibler Divergence in Knowledge Distillation for Large Language Models. COLING 2025.

[5] ABKD: Pursuing a Proper Allocation of the Probability Mass in Knowledge Distillation via alpha-beta-Divergence. ICML 2025.

[6] DA-KD: Difficulty-Aware Knowledge Distillation for Efficient Large Language Models. ICML 2025.

**Questions:**

In addition to the issues mentioned in the Weaknesses, I have a few more concerns:

1. Although the authors mention high computational cost, the paper lacks concrete efficiency analysis or runtime comparison experiments.
2. Is it always better to use more and stronger teacher models?

---

### Comment · Area_Chair_ouFG · 2025-11-22

Dear Reviewers,

Thank you for your time and effort in reviewing submissions for ICLR  2026. As we begin the author-reviewer discussion process, we kindly remind you to submit your responses to the author rebuttals by **December  2**.


Your engagement in this discussion phase is crucial to ensuring a fair and thorough evaluation of each submission.

**Action Required**


- Carefully consider the authors’ rebuttal and any additional evidence they provide.

- Update your review (if applicable) to reflect your revised perspective.

-  **Discuss with the authors if further details are required**


Your AC

---

### Meta-Review · Area_Chair_4sPA · 2026-01-06

**Summary:**

The reviewers raised several concerns about the clarity, novelty, and generalizability of the paper's approach. First, the motivation for using multiple drifting MLLMs for distillation was seen as unclear, particularly in terms of why a multi-teacher setup is necessary or more effective than traditional methods. The experiments were limited to a single medical domain (chest X-rays), which left the broader applicability of the method uncertain. Additionally, the assumptions made, such as the independence of teacher models’ reasoning trajectories, were viewed as unrealistic and overly simplistic. There were also significant concerns about the lack of baseline comparisons, which hindered the ability to assess the effectiveness of the proposed method. Furthermore, the theoretical contributions were deemed unclear, with confusing notation and a lack of formal definitions for key concepts like "concept drift". Given these issues, the reviewers expressed skepticism about the paper's overall contribution and recommended rejection.

**Reviewer Concerns:**

It appears that the authors did not submit a rebuttal. As a result, many of the reviewers' concerns, such as the clarity of the motivation, the generalizability of the approach, and the unrealistic assumptions made in the paper, remain unaddressed. The lack of rebuttal or response left reviewers uncertain about how the authors would clarify the key issues, such as the assumptions regarding the independence of teacher models and the methodology’s broader applicability beyond the medical domain.

**Reviewer Scores:**

Since the authors did not engage in the discussion phase, it's unlikely that the reviewers' scores would change. Most reviewers would maintain their initial assessment, with rejection being the most likely outcome.

---

### Decision · Program_Chairs · 2026-01-26

Reject